# Is Regulation Protection? Forest Logging Quota Impact on Forest Carbon Sinks in China

**Ziqiang Zhang** [1,*] **, Jie He** [2] **, Ming Huang** [1] **and Wei Zhou** [3]

1   School of Economics, Guizhou University, Guiyang 550025, China; gs.huangm22@gzu.edu.cn
2   School of Tourism and Cultural Industry, Guizhou University, Guiyang 550025, China; gs.jiehe21@gzu.edu.cn
3   College of Economics and Management, South China Agricultural University, Guangzhou 510642, China;
    zhouw@scau.edu.cn
*   Correspondence: zqzhang5@gzu.edu.cn

**Abstract:** As the central part of terrestrial ecosystems, forests have an irreplaceable role in regulating climate, prompting various efforts to protect them. Logging regulation is the most commonly used forest conservation strategy. Although the logging permit scheme was written into the Forest Law in China, its effect on forest carbon sequestration has rarely been subject to careful empirical scrutiny. In this paper, we develop and estimate a spatial econometric model to disentangle its potential effects on forest carbon sinks based on a panel dataset of 29 provinces from 1989 to 2018 in China. Our calculations show that China's forest carbon sinks are still growing and are connected geographically, with a tendency towards "high-high" and "low-low" aggregation. Increasing the logging quota produced a spatial spillover effect that might encourage the formation of forest carbon sinks in nearby areas. It considerably encouraged the expansion of forest carbon sinks. Additional mechanism testing is consistent with the claim that rising logging quotas have significantly boosted the proportion of timber forests in afforestation but had no effect on the movement of rural labor to urban areas. The development of forest carbon sinks is impacted in different ways by various logging quota types, with an increasing tendency for logging quotas to have a more significant contribution. Additionally, the non-collective forest region has a more significant spatial spillover effect of the logging quota on forest carbon sinks. The logging quota scheme should be improved by policymakers, beginning with eliminating tending quotas in the southern collective forest region. After that, the logging quota would gradually be eliminated nationwide, notably for commercial forests.

**Keywords:** regulation; logging quota; forest carbon sinks; spatial econometric model

## 1. Introduction

The forest carbon sink is an important way to achieve carbon (C) sequestration and requires the strengthening of forest protection and sustainable management [1]. The management of forests for carbon sequestration and the use of wood for energy relies on the functioning of forested ecosystems [2], which may lead to trade-offs. According to certain studies [3], there is a trade-off between the amount of carbon stored in current forests and the amount of timber that can be extracted from those forests. Such trade-offs do not exist in practice, and harvesting restrictions do not always increase forest sequestration [4].

As the most significant carbon sink and most cost-effective carbon absorber on land, forests are known for their high capacity as a carbon sink, which can effectively limit carbon dioxide production and other greenhouse gases [5]. China's average annual net growth of forest area over the previous 10 years has reached 1.9 million hectares, according to the Global Forest Resources Assessment 2020, making it the country with the most significant average annual net growth of forest area worldwide [6]. From 2009 to 2013, the amount of forest carbon sinks was 17.544 billion tons; from 2014 to 2018, it increased to 20.352 billion tons [7]. Because of this, some politicians and scholars have interpreted these modifications as proof that the logging restriction has been successful [8,9]. The primary idea behind

China's 1985 implementation of a logging quota system is that the amount of logging cannot outpace forest expansion. Any logging that occurs without a permit or that exceeds the allotted limit is, therefore, illegal and punishable [10].

This harvesting restriction, however, has generated debate ever since it was implemented [11]. Liu Shilei and Xia Jun [12] found that while the logging quota scheme has effectively halted the country's deforestation and forest degradation from the late 1980s to the early 1990s, it has played a minimal role in promoting forest restoration and regrowth by stifling investment in forest management. China has been planting and repairing forests, and new plants are good at absorbing $CO_2$. Improving forest management practices, instead of establishing new trees, does not call for a change in the land's cover, which prevents direct conflicts with the production of food [4]. The logging quota scheme was kept in place when forest law was revised in July 2020. It is reasonable to assume that the scheme will continue to impact forest carbon sinks for years. Therefore, analyzing its effectiveness and impact is pertinent and interesting.

This question will be empirically addressed as part of our investigation. In order to achieve this, we will construct and estimate a thorough causal model using a panel dataset at the province level. In various respects, this essay advances the knowledge of timber logging quotas. First, rather than focusing on the size of the logging quota, many studies primarily criticize the inefficiency of the quota scheme, which results in the ineffectiveness of forest protection. As a result, the quantity of the logging quota has not had a favorable impact on forest carbon sinks. This study further examines whether logging quota restrictions can be loosened by raising the quota's volume to encourage the development of forest carbon sinks. Secondly, this research employs a spatial econometric model to empirically assess the spatial spillover effect of the logging quota on forest carbon sinks, taking into account the externality of forest protection, to validate the spatial spillover effect of the logging quota scheme on forest carbon sinks. Third, even though the logging quota's impact on forest conservation has been extensively debated, more research has to be conducted on how it works. Due to the analysis of the impact of the logging quota on forest carbon sinks, this paper further tested its influence mechanism.

## 2. Literature Review

To increase the carbon storage capacity of forest ecosystems, countries have adopted a variety of forest protection policies and programs. The United States enacted the Forest Reserve Act in 1891 and has continuously adjusted its milestones and measures since then, combining forestry support policies and tax policies on a policy basis [13], and coordinating and guiding forestry affairs from the three perspectives of policies, incentives, and economic measures, thus serving as a model of forest governance for other nations. Developing nations are progressively recognizing the significance of forest environmental governance. There is evidence that measures such as the establishment of protected areas, payments for ecosystem services, and community-based management have been effective in expanding the forest cover in countries like Mexico [14]. Countries including Ecuador, the Philippines, and Zambia are also safeguarding their forests by reforesting and combating illegal logging [15]. What endeavors has China made to manage the forest environment? China has implemented a series of forestry policies including returning farmland to forests, natural forest protection, forest logging restrictions, improving collective forest rights, establishing protected areas, building national parks, etc., while building an ecological compensation mechanism with economic incentives such as subsidies, etc. [16]. As is well known, the effects of environmental regulation policies with regard to content and severity vary [17]. Among China's present forest management measures, the logging quota system is a more direct and powerful policy instrument; however, there are numerous controversies regarding the effects of this policy's implementation.

First, some studies have concluded that logging quota systems have no effect on the growth of forest carbon sinks. This is due to the logging quota scheme's prohibition on forest managers operating their timber output in a way that maximizes their profits as well

as the fact that it makes it less specific but less favorable for them to continue receiving advantages from forestry in the future [18]. The fact that it redistributes the tenure rights of forest land and forests to individual households has thus partially countered the better incentives for forest management and timber production resulting from the collective forest tenure reform [19]. Numerous empirical studies have demonstrated that while the logging quota scheme may have temporarily halted further deforestation and forest degradation, it has little to no long-term impact on the growth of forest carbon sinks. For instance, Xu et al. [20] found that higher logging regulations failed to restore the state-owned forests; instead, they caused a loss in forest growth over time, using panel data from 28 provinces. Jiang et al. [21]'s analysis of data from the northeast between 1980 and 2004, using a behavioral model for state-owned enterprises (SOEs), revealed that the logging quota scheme had no impact on safeguarding state forests. He et al. [22] used breakpoint regression on provincial panel data and discovered that the logging quota scheme did not encourage the increase of forest stock.

Second, other criticisms of the logging quota scheme also exist. One is that it incurs a significant financial and administrative burden. Local governments have been forced to employ many experts to supervise the entire timber collection, transportation, and distribution process to implement the ban [22]. Second, the logging quota scheme has significantly distorted the market and slowed the expansion of global forest carbon sinks by limiting the domestic timber supply. The logging quota scheme led to a significant shift in forest exploitation in Southeast Asian nations with lax regulatory systems and Russia [23]. Their findings show that big afforestation and reforestation programs, rather than the logging quota scheme itself, are primarily responsible for China's forest recovery and increased forest carbon sinks.

As a result, long-term forest carbon sink improvements could not be as significant as anticipated by policymakers. Therefore, it is essential to consider the effects of the logging quota scheme on forest carbon sinks in a more impartial, organized manner.

## 3. Theoretical Framework

### 3.1. The Logging Quota Scheme's Historical Context

In the late 1970s, the People's Communes were demolished, and in the 1980s, China's rural areas underwent reform. The Household Responsibility System (HRS), a name given to these reforms, expanded quickly [24]. Rural households wanted an equal chance in forestry after seeing the success of the HRS. To formalize both family forest management and the HRS inside the collectives, the State Council released "The Resolution on Several Issues Concerning Forest Protection and Development" in 1981, and the local authorities did implement a system like the HRS [19]. Additionally, all provinces and autonomous regions must strictly regulate the amount of logging because wood consumption is less than growth. Although the logging quota scheme was included in forest law in 1985, it still needed to provide concrete plans for putting the principle into practice [22]. It was not until 1991 that the logging quota scheme was introduced nationwide because of the incomplete forestry sector and the lack of staff at the time [22]. While this is happening, several institutional restraints either continue to limit some domestic activity or foster an uncertain environment. The logging quota system was kept in place when forest law was revised in July 2020.

### 3.2. Theoretical Analysis

Increasing the logging quota promotes the development of carbon sinks in the forest. From the perspective of property rights control, the logging quota scheme makes the management rights of farmers incomplete, and the unstable property rights will place the forestland at risk of being arbitrarily adjusted and expropriated, which reduces the incentive of farmers to plant forests [25]. Applying for logging targets increases the transaction costs between rural households and forestry department staff, which reduces their incentives for afforestation and hinders the development of forest carbon sinks [26]. In addition,

the forest ecosystem is an organic whole, and there is evidence that forest carbon sinks have cross-regional spatial correlation characteristics [27]. If the logging limit of a forest is increased, farmers will be more active in afforesting this area, which will increase the carbon sink of this forest while also benefiting the forest carbon sinks of the neighboring areas. Here, the impact of forests as carbon sinks are seen in multiple dimensions, including forest area, stocking volume, and efforts to establish new forests [28]. Additionally, there are two ways in which the logging quota scheme's effects on the different aspects of forest carbon sinks can be seen.

On the one hand, increasing the forest logging quota will encourage rural households to cultivate timber forests, thereby fostering the growth of forest carbon sinks. If the forest logging restriction is tightened, it will be more challenging for rural households who plant timber forests to reach their harvesting goals. If rural households' timber forests are classified as public welfare forests in the future, harvesting will be entirely prohibited and their losses will increase [29]. To avoid risk, rational rural households will convert their extant timber forests into economic forests [30]. If all rural households convert timber forests to economic forests, it will result in a single-species forest in one location, which is detrimental to forest ecosystem stability [31]. In addition, the carbon sequestration capacity of economic forests is frequently lower than that of timber forests [32], and the expansion of economic forests will reduce the carbon sequestration capacity of the forest area, which is detrimental to the growth of forest carbon sinks.

On the other hand, reducing the forest harvesting limit will not promote the growth of forest carbon sinks by causing rural labor to migrate to cities. It has been discovered that the policy of returning farmland to forests encourages rural households to seek non-agricultural employment [33], and that the restrictions on agricultural land imposed by China's Green Grain Project (GGP) also encourage the migration of rural labor [34]. In this context, the logging quota scheme for forest exploitation may play a similar function. The logging quota scheme has resulted in restrictions on the use of forestland [35], preventing rural households from earning a living wage from forest land operations and compelling them to relocate to the city. However, the practical difference is that, in general, rural households may make a living from agriculture but not forestry, and the logging quota has a limited impact on the livelihoods of rural households, who do not choose to go out to work as a consequence. Migration of rural labor is more likely to be influenced by urban employment opportunities and wages than by the logging quota. Consequently, the aforementioned impacts of forestry quotas may not exist.

Therefore, it is essential to consider the effects of the logging quota scheme on forest carbon sinks in a more impartial, organized manner. Another issue is that there may be spillover effects from the interplay between the logging quota plan and the forest carbon sinks. So, we chose to use spatial regression techniques in this study, which is a method of modeling issues related to natural resource use, theories of interacting agents, and interdependent decision making to deal with the potential biases embedded in the dynamic interaction and spillover response.

## 4. Materials and Methods

### 4.1. Date

Statistics on forests and tree crops were derived from the findings of the National Forest Inventories conducted between 1989 and 2018. And, the National Forest Inventory (NFI) is carried out every five years (between 1989 and 1993, 1994 and 1998, 1999 and 2003, 2004 and 2008, 2009 and 2013, and 2014 and 2018). Therefore, a corresponding sum over the entire reference period during which the inventory was conducted must be used for the period corresponding to each inventory rather than provincial figures for a single year [12,30]. National statistical yearbooks, national forestry statistical yearbooks, and national rural statistical yearbooks provide additional province-level data. Incomplete statistics existed for Shanghai's forest and tree crop industries, and Chongqing and Sichuan split in 1997. Except for Shanghai, Chongqing, Hong Kong, Macao, and Taiwan, we created

a longitudinal dataset of 29 provinces using these techniques for six inventory periods. Monetary variables are deflated and converted to 1988 constant prices using a GDP deflator or the consumer price index.

### 4.2. Method

The interrelationship of things strengthens with geographic closeness, according to Tobler's First Law of Geography [36]. The spatial distribution of woods exhibits significant variation, according to historical study. Additionally, it has been discovered that spatial spillover effects are maintained by forest carbon sinks [37]. A forest ecosystem is an organic whole. Deforestation on forestland impacts carbon sequestration inside the forestland itself and the carbon sequestration within the surrounding forestland.

### 4.2.1. Spatial Correlation Analysis

It is imperative to utilize spatial econometric approaches to account for estimation bias if the explanatory variables are spatially autocorrelated [38,39]. Specific measures, such as Moran's I and Geary's C index, were used to examine the global autocorrelation for complex natural phenomena. These indicators take into account the dependence and heterogeneity of spatial data. The global Moran's I index value ranges from −1.0 to 1.0. The stronger the spatial correlation, the higher the value of Moran's I; when Moran's I is 0, it indicates that there is no spatial correlation, making it inappropriate to use spatial econometric methods. When Moran's I is greater than 0, it indicates that y has spatial positive correlation; when Moran's I is less than 0, it indicates that y has spatial negative correlation. Additionally, the contiguity weight matrix $W_1$, distance weight matrix $W_2$, and economic weight matrix $W_3$ were chosen for spatial econometric analyses, for which the calculations are shown in Equations (1)–(4), in order to more accurately reflect the characteristics of spatial dependence and test the robustness of the research results.

$$W_1 = W_{ij} = \begin{cases} 1 \ if \ i \ and \ j \ are \ contiguity \\ 0 \qquad otherwise \end{cases} \tag{1}$$

$$W_2 = W_{ij} = {}^1\!/_{d_{ij}} \tag{2}$$

$$W_3 = W_{ij} = \frac{1/\left|\overline{E_i} - \overline{E_j}\right|}{\sum\limits_{j \in J_i} 1/\left|\overline{E_i} - \overline{E_j}\right|} \tag{3}$$

$$\overline{E_i} = (1/t_n - t_0 + 1)\sum\nolimits_{i=t_{it}}^{t_{it}} E_{it} \tag{4}$$

where $W_{ij}$ is the spatial weight matrix in spatial unit $i$ and $j$ that measures the strength of the relationship between two spatial units; $d_{ij}$ is the Euclidean distance from the government seat of province $i$ to that of province $j$; $t_n$ is the end of the study; $t_0$ is the base period of the study; $E_{it}$ is the real GDP per capita of province $i$ in year $t$; $J_i$ is the set of provinces other than $i$.

### 4.2.2. Spatial Econometric Model

Spatial econometric models can efficiently address the geographical dependencies missed by linear regression studies. The spatial Durbin model (SDM), spatial autoregressive model (SAR), and spatial error model (SEM) are the most often employed spatial panel models. Equations (5)–(7), respectively, are the generic forms of the SDM, SAR, and SEM:

SDM:
$$Y = \rho WY + \beta X + \lambda WX + \varepsilon \tag{5}$$

SAR:
$$Y = \rho WY + \beta X + \varepsilon \tag{6}$$

SEM:

$$\begin{cases} Y = \beta X + \varepsilon \\ \varepsilon = \lambda W \varepsilon + \mu \end{cases} \tag{7}$$

where $Y$ means the vector of response variable, $X$ means the matrix of n × k independent predictors, $\beta$ reflects the coefficient matrix of $X$, $\varepsilon$ means the vector of random disturbance term, $W \times \varepsilon$ means the spatial lag of $\varepsilon$, $\mu$ means the vector of random error term under normal distribution, $\rho$ means the coefficients of spatial regression terms, $W$ means then n×n spatial weight matrix, $W \times Y$ means the spatial-lag response variable, $\lambda$ means the coefficients of spatial random error terms for the vector of cross-sectional response variable.

Following Liu et al. [12], we must eliminate models through a first difference of spatial econometric models to resolve these problems. This is Equation (8), (using the SDM model as an illustration):

$$\Delta Y = \rho W \times \Delta Y + \beta \times \Delta X + \lambda W \times \Delta X + \Delta \varepsilon \tag{8}$$

where $\Delta$ is the operator for making first differences, the direct effects of independent variables refer to the influence of regional influencing factors on the regional forest carbon sinks, and the indirect effects (i.e., spatial spillover) can be explained in two ways, by regional influencing factors on the regional forest carbon sinks and by regional influencing factors on the province of interest. The SAR and SDM can estimate independent variables' direct and indirect effects.

### 4.3. Variable Definition
### 4.3.1. Outcome Variable

The outcome variable was decided upon as forest carbon sinks. Most approaches for calculating forest carbon sinks rely on microscopic monitoring, which is challenging. These methods include the biomass, carbon density, and carbon balance methods. The more popular method with good operability and applicability is forest stock expansion. The general development trend of forest carbon sinks can be reflected by this measurement approach, even though it cannot precisely estimate the storage of forest carbon sinks. The following is the formula for calculating the forest stock expansion technique.

$$TC_i = (1 + \alpha + \theta)V_i \delta \sigma \gamma \tag{9}$$

$$V_i = \sum_{j=1}^{n} (S_{ij} \times V_{ij}) \tag{10}$$

where $TC_i$ is the carbon storage of the forest ecosystem in province $i$; $S_{ij}$ is the area of the ith-type tree species in province $i$(ha); $V_{ij}$ is the storage volume per unit area of the $i$th-type forest in province $i$ (m$^3$·ha$^{-2}$); $V_i$ is the forest stock in province $i$(t C); coefficients in Equation (9) are determined by default parameter values proposed by IPCC (International Plant Protection Convention). $\delta$ is the coefficient of the biomass stocks converted by forest stock volume (also called the biomass expansion coefficient). According to the IPCC's recommendations, it is derived from the relationship between biomass and forest stock volume and is generally taken as 1.9 [40], dimensionless parameters. $\sigma$ is the C density coefficient of the forest biomass stocks converted into dry biomass; it is closely related to forest stock volume. IPCC listed this common conversion factor as 0.45–0.50; 0.5 was used here. $\gamma$ is the proportion of carbon in the dry biomass, i.e., the coefficient of carbon sequestration converted by dry biomass. On average, 50% of the biomass is estimated as the carbon content for all species of trees [41]; the parameter data were calculated by Zhang et al. [42] and they used 0.5. $\alpha$ is the carbon conversion coefficient of forest land, and its value is 0.195. $\theta$ is the carbon conversion coefficient of forest land and its value is 1.244.

### 4.3.2. Explanatory Variable

The explanatory variable used was the logging quota. Although the logging quota scheme was included in forest law in 1985, no concrete plans for its implementation were provided until 1991. According to the five-year plan, China introduced the logging quota scheme. The state approves total logging quotas for five years, not annually. An NFI is additionally conducted every five years in China. Data collection lasts for the same amount of time, but it begins and ends in different years. The data on logging quotas was initially gathered in China between 1991 and 1995 as part of the eighth five-year plan. Data on forest resources for a comparable time frame were gathered from the 4th NFI (1989–1993). Therefore, the average logging quota of the two subsequent five-year plan periods was used for processing to correct the data collection cycle's misalignment. The state-approved logging quota is divided into three categories: primary logging, tending logging, and other logging. The difference processing of the logging quota is no longer carried out since the logging quota scheme is the external limitation imposed on the protection of the forest, and the logging quota is statistical data with a 5-year cycle.

### 4.3.3. Variables Used in Mechanism Test

We conducted additional tests to evaluate the theories that account for the impacts of the logging quota on adjustments to forest carbon sinks. It still needs to be discovered how the alternative livelihood effect works. For this, we employed a model with the same composition as Equation (8). On the one hand, it has been proposed that the logging quota scheme might free up rural households for off-farm work and encourage rural labor to flow to cities, easing the strain on forestland and fostering an increase in forest carbon sinks. To indicate the rate of rural labor flow, we created the following variable:

$$Tr = \frac{R_L - R_N}{R_L} \tag{11}$$

where $R_L$ is the number of laborers in rural areas, $Tr$ is the rate of rural labor migration, and $R_N$ is the number of laborers employed in agriculture, forestry, animal husbandry, and fisheries. The rate of rural labor flow is more significant when the $Tr$ is higher, and vice versa. In order to indicate the intensity of rural labor flow to urban areas, we employ the variable. The logging quota scheme, on the other hand, prohibits forest managers from managing their timber production in a way that maximizes their profits [18]. Rural households frequently overlook the proper planting under the logging quota regime, which results in significant land use defects.

Rural households in China that grow wood forests run the risk of having their future access to logging rights restricted due to the country's tightening of environmental regulations. This could lead to a prompt reduction in the amount of timber forest afforestation, which would be detrimental to the development of forest carbon sinks. To gauge the amount of replanting of forests for timber, we created the following variable:

$$Aff = \frac{T_f}{Z_f} \tag{12}$$

where $T_f$ is the amount of timber forest afforestation during the NFI period, $Z_f$ is the amount of total afforestation during the NFI period, and Aff is the percentage of timber forest afforestation area in the total afforestation area. It serves to tally the area planted with trees annually during the NFI period. The afforestation area is not considered different because it is a net change amount.

### 4.3.4. Control Variables

Conceptually, the extent to which policies and programs have been implemented and other social-ecological factors affect forest carbon sinks [19]. The control variables included in this study's list of potentially affecting factors (together with an associated list of descriptive data shown in Table 1) are as follows.

**Table 1.** Definition, assignment, and statistical description of variables that may affect forest carbon sinks.

| Variables | Description | Mean | Std. Dev. | Min | Max |
|---|---|---|---|---|---|
| Outcome variables | | | | | |
| Carbon | Forest carbon sink variation over two successive NFI periods (million tons) | 0.612 | 1.182 | −1.705 | 11.732 |
| Explanatory Variables | | | | | |
| Quota | Total logging quota, million cubic meters | 8.285 | 8.784 | 0.020 | 40.714 |
| $Quota_1$ | Principal logging quota, million cubic meters | 3.712 | 4.962 | 0.000 | 36.708 |
| $Quota_2$ | Tending logging quota, million cubic meters | 1.685 | 1.753 | 0.000 | 9.620 |
| $Quota_3$ | Other logging quota, million cubic meters | 2.728 | 3.643 | 0.000 | 21.646 |
| Variables used in mechanism test | | | | | |
| Mobility | Rural labor force migration rate changes (%) | 0.036 | 0.132 | −0.440 | 0.350 |
| Timber | The proportion of timber forest afforestation area in total afforestation area (%) | 0.286 | 0.213 | 0.089 | 0.875 |
| Control variables | | | | | |
| Pgdp | The change in GDP per capita over two successive NFI periods, CNY10,000, log form | 0.395 | 0.187 | −0.387 | 1.063 |
| Income | The change of the average per capita net income of rural households over two successive NFI periods, CNY, log form | 0.423 | 0.362 | −0.204 | 1.365 |
| Urban | The change of urban population proportion over two successive NFI periods, % | 4.740 | 4.924 | −3.451 | 32.597 |
| Grain | The change of grain sown area over two successive NFI periods, thousand ha, log form | −0.023 | 0.179 | −1.099 | 0.472 |
| Reform | Assign the reform year and subsequent years a value of 1, otherwise 0 | 0.454 | 0.499 | 0.000 | 1.000 |
| Closure | Assign 0.3 to 0, otherwise 1 | 0.833 | 0.374 | 0.000 | 1.000 |
| Time | The 4th NFI is assigned a value of 1, the 5th NFI is assigned a value of 2, and so on to the 9th NFI is assigned a value of 6. | 3.500 | 1.713 | 1.000 | 6.000 |

(1) GDP (Gross Domestic Product) per capita: Macroeconomic elements are represented by GDP per capita. According to research on forest transition, economic growth may encourage off-farm migration and the abandonment of marginal land, opening up prospects for forest regeneration [43]. The relationship between forest acreage and GDP per capita is another topic of study for environmental Kuznets curves (EKC) for deforestation [44]. The Environmental Kuznets curve shows that as development occurs, pollution first increases and then decreases because people value clean air.

(2)     The income of rural households: According to Yan et al. [30] the cultivation of tree crops may offer disadvantaged households an alternative source of income. Compared to other rural residents, disadvantaged rural people may rely more heavily on income from the primary sector. The share of primary sector income in the overall income of rural households has declined [45]. Off-farm work provides disposable income for rural households, not growing tree crops. Therefore, the dependency on the mountain forest decreases as rural household disposable income increases. As a result, the need for logging by rural households to support themselves decreases, which is good for conserving the forest's ecology.

(3)     Urbanization: 35.28% of China's population still resides in rural areas, and there needs to be more forested land to accommodate the country's enormous population [46]. Forest management is evolving in post-industrial cultures to prioritize amenities over timber production [47]. Higher levels of urbanization increase the likelihood that agricultural land will become marginalized increase the likelihood of forest restoration and transformation, and impact forest carbon sinks. The percentage of the population living in urban areas at the end of the year is used to gauge urbanization levels.

(4)     Grain planting scale: Forestland is frequently transformed into agricultural areas for dietary needs after logging. Due to competition with forestry output for land usage, agricultural production is a significant factor in the decline of forestland [48]. The size of grain planting directly impacts forest carbon sinks. Grain planting area serves as a proxy for grain planting scale.

(5)     Collective forest tenure reform. State and collective ownership are China's two primary forms of forestland ownership. China has been implementing collective forest tenure reform since 2003, often known as the "new round of forest reform", based on unchanging collective ownership. The current cycle of forest reform has been promoted over the entire nation after being tested in Fujian, Jiangxi, Yunnan, and other locations. The reform's implementation date differs from province to province. With the establishment of forestry business entities and the activation of forestry businesses, forest land, and forest tenure rights are reassigned to specific households [49]. To distinguish the impact of the reform, we include a dummy variable [19]. Assign a value of 1, otherwise, to the reform year and all succeeding years.

(6)     Canopy density: In the 5th NFI, Chinese forestry officials changed the minimum 30% tree crown cover requirement to 20%. We add a dummy variable to identify how this modification will affect things [30].

(7)     Trends in time: The NFI is conducted every five years. Time effects must therefore be considered for the period corresponding to each inventory rather than for local numbers for a single year. Give the stage of forest inventory a value. According to Yan et al. [30], the fourth NFI is given a value of 1, the fifth NFI is given a value of 2, and so on, until the ninth NFI is given a value of 6.

This study explored the logarithm values of serval control variables to normalize the data for enhanced comparability. Table 1 primarily serves as a visual representation of the variables and descriptive statistical analyses that may affect the forest carbon sink.

As illustrated in Figure 1, it is created using Excel 2010 software based on the NFI data; China's logging quota and forest carbon sinks increased significantly during the past forty years, whereas the total quota nearly remained constant. Each NFI period saw a significant increase in the shift of forest carbon sinks. The increase in forest carbon sinks during the ninth NFI, to 2.807 billion tons, was the greatest among them. The logging quota tends to stabilize at around 250 million cubic meters; the central logging quota and tending logging quota both exhibit a slight increasing tendency, while the other logging quota exhibits a clear decreasing trend.

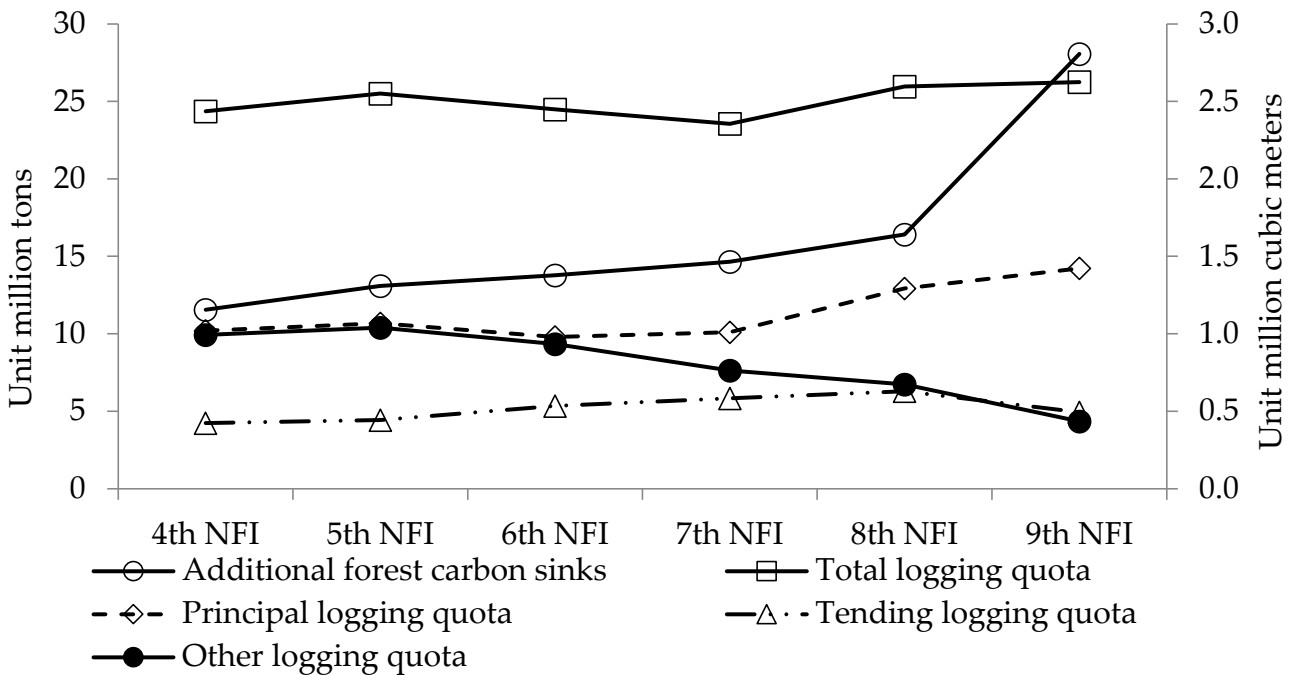

**Figure 1.** Growth trends for forests as carbon sinks and logging quota.

## 5. Results

### 5.1. Spatial Effects

Using a set of contiguous data, namely $W_1$, the Global Moran's I index was utilized to examine the spatial association between forest carbon sinks and the logging quota across the 29 provinces. The findings demonstrate that the I value for logging quotas is all positively significant under the $W_1$ during the analysis years, and then I value for forest carbon sinks is positively significant from the 6th NFI, demonstrating a considerable positive spatial autocorrelation. The Local Moran's I index maps the scatter plot (Figure 2) and shows the spatial autocorrelation patterning. As illustrated in Figure 2, it is created using Stata15.0 software based on the NFI data; most areas exhibited a coexistence of both high–high and low–low types. The findings supported the global spatial correlation test by demonstrating that forest carbon sinks exhibited spatial autocorrelation.

### 5.2. Analysis of Spatial Panel Regressions

The Wald test and likelihood ratio (LR) test are employed for model selection in this study because it is hard to verify the existence or form of the spatial correlations [50]. Additionally, Hausman's test recommends using random-effects estimates. The three spatial weight matrices are significant under the three spatial weight matrices, indicating that they could not be successfully degenerated into SEM or SAR models, according to the Wald and LR tests (Table 2). The empirical findings of the SDM model for random effects are the main focus of the remaining portions of this study.

According to the regression results (Table 3), there is a significant spatial association and positive spillover effect of forest carbon sinks between provinces. The spatial autoregressive coefficient rho values are positive and significant at the 5% level under the three spatial weight matrices. The logging quota's influence coefficients on forest carbon sinks are noticeably positive, indicating that tightening logging management by lowering the quota is not advantageous to developing forest carbon sinks. According to one probable explanation, the logging quota may diminish forestry management expectations and lead to short-term behavior. The amount of time and effort rural households expend to collect the logging index lowers their excitement for afforestation and tending and affects the society's enthusiasm for investing in forestry [18]. However, under the influence of economic interests, illegal timber logging may be encouraged [51], which is detrimental to expanding

forest carbon sinks. Although the logging quota can, to some extent, encourage the protection of forest resources, the growth effect of forests as carbon sinks cannot outweigh this effect. Therefore, tightening logging regulations will not generally promote the expansion of forest carbon sinks.

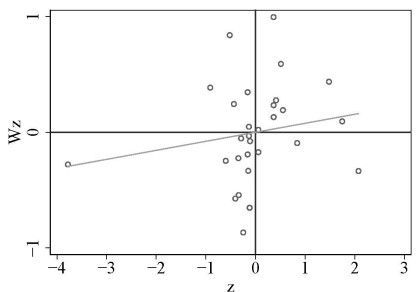

(**a**) Moran's I index of forest carbon sinks based on the 4th NFI data

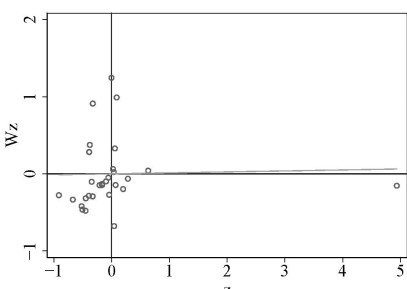

(**b**) Moran's I index of forest carbon sinks based on the 5th NFI data

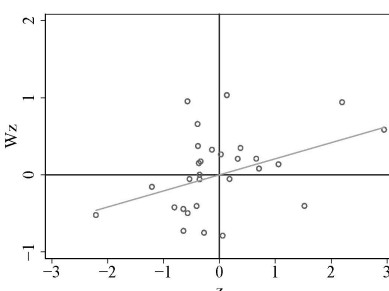

(**c**) Moran's I index of forest carbon sinks based on the 6th NFI data

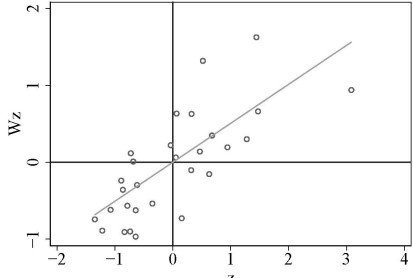

(**d**) Moran's I index of forest carbon sinks based on the 7th NFI data

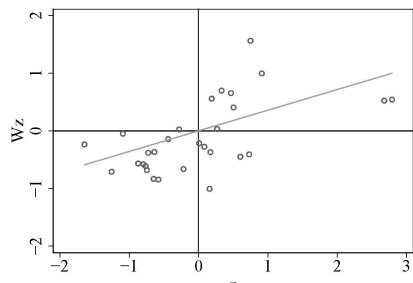

(**e**) Moran's I index of forest carbon sinks based on the 8th NFI data

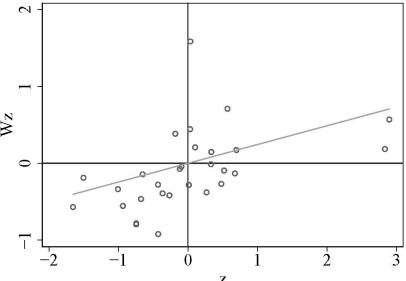

(**f**) Moran's I index of forest carbon sinks based on the 9th NFI data

**Figure 2.** Local Moran's I index scatter plot of forest carbon sinks.

**Table 2.** Test results of spatial econometric models.

| Spatial Weight Matrix | Test Method | SAR | SEM | SDM |
|---|---|---|---|---|
| $W_1$ | Wald test | 28.00 *** | 25.07 *** | |
| | LR test | 26.06 *** | 26.01 *** | |
| | Hausman test | | | −7.03 |
| $W_2$ | Wald test | 2854.30 *** | 35.56 *** | |
| | LR test | 23.54 *** | 23.65 *** | |
| | Hausman test | | | −7.35 |
| $W_3$ | Wald test | 4759.10 *** | 25.13 *** | |
| | LR test | 18.60 *** | 18.68 *** | |
| | Hausman test | | | 2.56 |

Notes: Parenthetical values are t-statistics. *** represent significance at the 1% levels, respectively. Values in parentheses are standard errors.

**Table 3.** Estimation of spatial effects of the logging quota on forest carbon sinks.

| Variables | Carbon | | |
|---|---|---|---|
| | $W_1$ | $W_2$ | $W_3$ |
| Quota | 0.022 ** (0.012) | 0.024 ** (0.011) | 0.028 ** (0.012) |
| Pgdp | 0.269(0.571) | 0.176 (0.556) | −0.196 (0.514) |
| Income | 2.476 *** (0.726) | 3.166 *** (0.760) | 2.750 *** (0.728) |
| Urban | 0.029 ** (0.014) | 0.023 * (0.013) | 0.030 ** (0.014) |
| Grain | 0.726 (0.538) | 0.897 * (0.527) | 0.542 (0.546) |
| Reform | 0.352 (0.512) | 0.074 (0.477) | −0.021 (0.396) |
| Closure | 0.039 (0.000) | 0.352 (0.000) | −0.206 (0.602) |
| Time | 0.042 (0.000) | −0.083 (0.535) | 0.235 (0.330) |
| $W \times$ Quota | 0.025 * (0.019) | 0.039 * (0.026) | −0.013 (0.047) |
| $W \times$ Pgdp | −0.551 (0.845) | −1.512 * (1.040) | −0.179 (1.096) |
| $W \times$ Income | −2.624 *** (0.757) | −3.403 *** (0.826) | −2.950 *** (0.811) |
| $W \times$ Urban | −0.019 (0.027) | 0.015 (0.024) | 0.005 (0.032) |
| $W \times$ Grain | 0.154 (0.964) | −0.049 (0.814) | 1.409 (1.316) |
| $W \times$ Reform | −0.766 (0.821) | 0.162 (0.944) | −0.712 (0.947) |
| $W \times$ Closure | −0.009 (0.463) | 0.107 (0.525) | −0.023 (0.000) |
| $W \times$ Time | 0.123 (0.203) | 0.007 (0.000) | 0.027 (0.000) |
| rho | 0.004 ** (0.109) | 0.043 ** (0.179) | 0.059 (0.156) |
| sigma2_e | 1.071 *** (0.130) | 1.030 *** (0.123) | 1.013 *** (0.124) |
| $R^2$ | 0.214 | 0.225 | 0.201 |
| Direct effect | 0.023 ** (0.012) | 0.024 ** (0.012) | 0.028 ** (0.013) |
| Indirect effect | 0.026 * (0.021) | 0.037 * (0.026) | −0.015 (0.044) |
| Total effect | 0.049 *** (0.017) | 0.061 *** (0.022) | 0.013 (0.050) |

Notes: Parenthetical values are t-statistics. ***, **, * represent significance at the 1%, 5%, and 10% levels, respectively. Values in parentheses are standard errors.

Under the three spatial weight matrices, the control variables, rural household income and urbanization have both shown statistically significant positive effects on forest carbon sinks, suggesting that raising rural household income and urbanization levels helps foster the expansion of forest carbon sinks. Here, is one explanation: as rural households' earnings rise, particularly their off-farm earnings, their dependence on forestland falls [30], and the likelihood of illegal logging is lower. In general, the less rural households have to rely on forestry to support their livelihood, the more significant their income. Besides, a significant portion of the rural labor force also leaves the countryside as urbanization levels rise and the aging of the rural population quickens. Rural households' willingness to use their forestland as public welfare forests is increased due to the labor shortage in forestry planting. So, they will not overcut trees or engage in illegal logging, promoting the expansion of forest carbon sinks. At the same time, they are more cognizant of the need to protect forests. Ecotourism is a sustainable use of forest resources compared to traditional services such as logging. Besides, ecotourism can protect forests when accompanied by conservation mechanisms (e.g., protected area, payment for ecosystem services, monitoring/enforcement) [52]. Ecotourism allows residents and governments to generate tourism income without consuming forest resources. Additionally, the marginalization of agricultural land encourages the conversion of farmland to forest [53], and forest transformation encourages the expansion of forest carbon sinks. Urbanization also reduces the conflict between people and land in rural areas. Only under $W_2$ does grain size have a considerable

beneficial impact on forest carbon sinks, indicating that increasing grain size encourages the development of these sinks. Increasing the planting scale in significant grain-producing regions reduces the strain on nearby grain production. It prevents competition between agricultural and forested areas, encouraging the conversion of farmland to forests and the transformation of forests.

The spatial lag term of the logging quota's coefficients ($W_1 \times$ Quota and $W_2 \times$ Quota) is highly positive, showing that the logging quota in the region significantly promotes the regional forest carbon sinks. The potential explanation is that raising the quota for logging in the surrounding areas can improve the expectation of forestry management and the security of forestland rights, attracting social investment [19]. Increased domestic wood production will reduce the discrepancy between local wood supply and demand. It lessens the possibility of illicit logging. Additionally, loosening logging restrictions in the vicinity can optimize the structure of newly planted forests, increase the stability of the entire forest ecosystem, and encourage the expansion of forest carbon sinks.

If there are spatial spillover effects, changes in some influencing factors will affect neighboring counties' forest carbon sinks and the local ones. This study divided the estimated results into total, direct, and indirect impacts to further analyze this spatial interaction. Under the $W_1$ and the $W_2$, the logging quota's direct, indirect, and overall effects on forest carbon sinks were noteworthy. As a result, the higher logging quota enhanced regional forest carbon sinks and those in nearby areas. In other words, the geographical spillover effect is the significant impact of the increased logging limit, which is also consistent with the SDM's results estimates. Therefore, easing logging regulations will help stabilize forestry management expectations, resolve the conflict between domestic wood supply and demand, and balance the economic and environmental advantages. The logging quota may have shifted forest harvest from places with one form of protection to another [54] rather than increasing forest protection. It can lessen the regulatory pressure on neighboring areas to conserve their forests, and the overflow of ecological advantages can encourage the expansion of carbon sinks in neighboring forests.

*5.3. Tests of Robustness*

Under the $W_3$, the logging quota's direct effects and overall impact on forest carbon sinks were insignificant. Thus, $W_1$ and $W_2$ are used to carry out the robustness test of the spatial effect. Results of robustness testing using two different types of variable choice are shown in Table 4.

The first is to replace forest carbon sinks through forest stock. The second is to directly replace the logging quota in each Chinese five-year planning term as a new explanatory variable (New Quota) in the SDM for estimation rather than averaging quota data. Instead of averaging the logging quota data, a third test that averages the growth of forest carbon sinks is carried out. The fourth is to swap out spatial regression for ols regression. Then, insert the new outcome variable (New Carbon) into the SDM for estimation. Test findings show that the conclusions made above are substantial and that the favorable impact of an increase in the logging limit on the expansion of forest carbon sinks continues to be statistically significant. The spatial lag term of the logging quota ($W_1 \times$ quota and $W_2 \times$ quota) shows strongly positive coefficients, demonstrating that the positive impact of raising the logging quota on forest carbon sinks continues to impact spatial patterns.

**Table 4.** Robustness tests: regression models testing effects of the logging quota on growth of forest carbon sinks.

| Variables | Forest Stock | | Carbon | | New Carbon | | Carbon |
|---|---|---|---|---|---|---|---|
| | $W_1$ | $W_2$ | $W_1$ | $W_2$ | $W_1$ | $W_2$ | |
| Quota | 0.019 * (0.010) | 0.021 ** (0.010) | | | 0.020 ** (0.010) | 0.020 ** (0.010) | 0.034 *** (0.010) |
| New Quota | | | 0.028 ** (0.011) | 0.028 *** (0.011) | | | |
| Pgdp | 0.233 (0.493) | 0.152 (0.480) | 0.246 (0.570) | 0.169 (0.554) | −0.267 (0.545) | −0.601 (0.530) | −0.353 (0.532) |
| Income | 2.138 *** (0.631) | 2.734 *** (0.656) | 2.515 *** (0.725) | 3.164 *** (0.754) | 1.358 ** (0.651) | 1.654 ** (0.718) | 0.249 * (0.327) |
| Urban | 0.025 ** (0.012) | 0.020 * (0.012) | 0.029 ** (0.014) | 0.023 * (0.013) | 0.015 (0.014) | 0.011 (0.013) | 0.033 ** (0.013) |
| Grain | 0.627 (0.464) | 0.775 * (0.456) | 0.714 (0.539) | 0.911 * (0.528) | 0.439 (0433) | 0.409 (0.424) | 0.552 (0.514) |
| Reform | 0.305 (0.443) | 0.064 (0.412) | 0.320 (0.513) | 0.042 (0.475) | 0.231 (0.293) | 0.207 (0.269) | −0.220 (0.398) |
| Closure | 0.033 (0.000) | 0.304 (0.000) | 0.025 (0.000) | 0.293 (0.524) | 0.047 (0.000) | 0.232 (0.352) | −0.199 (0.393) |
| Time | 0.036 (0.000) | −0.072 (0.205) | 0.047 (0.000) | −0.038 (0.239) | −0.003 (0.000) | 0.038 (0.117) | 0.077 (0.136) |
| $W \times$ Quota | 0.022 * (0.016) | 0.034 * (0.023) | 0.012 * (0.019) | 0.030 * (0.026) | 0.013 * (0.015) | 0.026 * (0.021) | |
| $W \times$ Pgdp | −0.475 (0.719) | −1.306 (0.898) | −0.559 (0.831) | −1.595 (1.036) | −0.325 (0.708) | −0.098 (0.957) | |
| $W \times$ Income | −2.267 *** (0.658) | −2.938 *** (0.713) | −2.673 *** (0.758) | −3.387 *** (0.820) | −1.760 ** (0.729) | −2.033 ** (0.863) | |
| $W \times$ Urban | −0.016 (0.018) | 0.013 (0.021) | −0.018 (0.021) | 0.017 (0.024) | −0.017 (0.022) | 0.002 (0.029) | |
| $W \times$ Grain | 0.134 (0.832) | −0.042 (0.702) | 0.165 (0.965) | −0.030 (0.812) | 0.504 (0.792) | 0.571 (0.731) | |
| $W \times$ Reform | −0.664 (0.706) | 0.141 (0.815) | −0.829 (0.824) | 0.011 (0.947) | −0.198 (0.501) | −0.290 (0.572) | |
| $W \times$ Closure | −0.008 (0.406) | 0.093 (0.453) | −0.027 (0.472) | 0.109 (0.000) | 0.282 (0.300) | −0.010 (0.000) | |
| $W \times$ Time | 0.106 (0.172) | 0.006 (0.000) | 0.142 (0.203) | 0.008 (0.000) | 0.021 (0.100) | 0.009 (0.000) | |
| rho | 0.033 ** (0.110) | −0.038 * (0.179) | 0.042 ** (0.109) | 0.033 * (0.179) | 0.073 ** (0.109) | 0.061 * (0.179) | |
| sigma2_e | 0.061 * (0.179) | 0.767 *** (0.091) | 1.062 *** (0.129) | 1.020 *** (0.122) | 0.311 *** (0.038) | 0.305 *** (0.037) | |
| $R^2$ | 0.214 | 0.225 | 0.215 | 0.229 | 0.303 | 0.282 | 0.241 |

Notes: Parenthetical values are t-statistics. ***, **, * represent significance at the 1%, 5%, and 10% levels, respectively. Values in parentheses are standard errors.

*5.4. Mechanism Test*

Rural households will consequently look for two alternative livelihoods because logging is regulated. The first is to convert the current wood forest into an economic forest in order to modify the species composition of the forest, because rural households typically need to convert the current timber forest into economic forest to avoid the loss of logging

that may be restricted in the future. Whatever the case, it should be noted that without a profit from logging, people would be unlikely to be motivated to plant more trees and exert greater maintenance efforts [18]. And the second is to look for off-farm employment in cities and towns. Based on the Grain for Green Project's (GGP) consequences in China, the use of agricultural land is also constrained. GGP has facilitated the transfer of labor forces and encouraged the transfer of other labor forces, except for those in rural areas, year by year [55]. Thus, it is necessary to take that possibility into account. As illustrated in Table 5, the logging quotas may influence forest carbon sinks through modifications to the species composition of the forest or rural labor migration. We investigated the mechanism for the outcome variables of rural labor migration and the percentage of timber forest afforestation area.

The logging quota significantly impacts the percentage of timber forest afforestation area, according to Table 5's results of the mechanism test under the $W_1$ and $W_2$ scenarios. At the same time, rural labor migration is not significantly impacted. The findings demonstrate that tighter logging regulations may result in a monoculture of forest species, which is harmful to the resilience of forest ecosystems. It is important to keep in mind that afforestation of a single species typically consumes a lot of soil nutrients and water, which reduces species abundance and biomass under forests [56]. The majority of plantation forests are made up of a single species of tree, which frequently has an adverse effect on the area's biological environment [31]. As a result, lowering the logging quota will not help forests become more effective carbon sinks. The reason is that while planting timber can aid in addressing the rural labor shortage and providing alternative sources of income, the logging quota alters rural households' planting habits, which could result in the planting of economically advantageous forests and alter the composition of the forest's tree species. The decrease in the scale of afforestation of timber forests does not support the development of forest carbon sinks since the carbon storage of timber forests in commercial forests is more significant than that of economic forests. Furthermore, the motivation for the logging quota needs to be improved, making it challenging to encourage rural households to migrate and lessen their reliance on forestland. This is because rural labor migration to urban areas strongly depends on non-farm employment opportunities. This further supports the analysis' findings that the logging quota does not promote the conservation of the forest's natural system.

*5.5. Spatial Heterogeneity Analysis*

The observation also accounts for the potential regional heterogeneity of logging quota impacts on $W_1$ and $W_2$ forest carbon sinks. The following two factors can be used to identify heterogeneity: On the one hand, the classification of the many logging quota categories. The outcomes of the logging quota structure are presented in Table 6 as heterogeneity. Only the lag coefficient of tending logging quota and other logging quotas ($W \times$ Quota) are significantly positive among the three types of logging quota's effects on forest carbon sinks.

The findings demonstrate that raising other logging quotas is advantageous for encouraging the development of forest carbon sinks. Raising the principal logging quota has no regional spillover effect. Increased tending and other logging quotas have a spatial spillover effect that encourages the expansion of nearby forest carbon sinks. The impact on the forest environment is more significant due to increased primary logging intensity. There is no spatial spillover effect even though the relaxed major logging limitation encourages social investment in forestry because large-scale logging will negate the forest's overall ability to store carbon [57]. The quality of forest management can be enhanced by attending to logging, such as removing disease-prone or dead trees and engaging in sanitary logging [16]. Therefore, lowering the tending logging quota can enhance the stand's quality, maximize its species and age composition, increase the forest ecosystem's stability, and consequently impact the surrounding space.

**Table 5.** Mechanism test of the effect of the logging quota on forest carbon sinks.

| Variables | Timber | | Mobility | |
|---|---|---|---|---|
| | $W_1$ | $W_2$ | $W_1$ | $W_2$ |
| Quota | 0.007 *** (0.002) | 0.008 *** (0.002) | 0.001 (0.001) | 0.001 (0.001) |
| Pgdp | 0.011 (0.053) | 0.009 (0.052) | 0.014 (0.034) | −0.021 (0.032) |
| Income | −0.084 ** (0.064) | −0.092 * (0.068) | −0.019 (0.040) | −0.015 (0.041) |
| Urban | 0.001 * (0.001) | 0.001 * (0.001) | 0.000 (0.001) | 0.000 (0.000) |
| Grain | 0.005 (0.055) | −0.026 (0.055) | −0.036 (0.032) | −0.044 (0.030) |
| Reform | −0.061 (0.049) | −0.028 (0.045) | −0.050 * (0.031) | −0.042 (0.027) |
| Closure | 0.004 (0.000) | −0.009 (0.048) | −0.017 (0.000) | 0.050 (0.000) |
| Time | 0.005 (0.000) | −0.010 (0.022) | 0.005 (0.000) | −0.014 (0.014) |
| $W \times$ Quota | 0.004 * (0.003) | 0.004 * (0.004) | −0.002 ** (0.001) | −0.004 *** (0.001) |
| $W \times$ Pgdp | −0.076 (0.077) | −0.056 (0.098) | 0.180 *** (0.053) | 0.319 *** (0.073) |
| $W \times$ Income | 0.035 * (0.068) | 0.044 * (0.075) | −0.158 *** (0.046) | −0.197 *** (0.053) |
| $W \times$ Urban | −0.006 *** (0.002) | −0.010 *** (0.002) | 0.000 (0.001) | 0.001 (0.001) |
| $W \times$ Grain | −0.096 (0.096) | −0.055 (0.080) | 0.078 (0.056) | 0.136 *** (0.048) |
| $W \times$ Reform | 0.095 (0.077) | 0.028 (0.089) | 0.126 ** (0.050) | 0.130 ** (0.056) |
| $W \times$ Closure | −0.020 (0.044) | 0.002 (0.000) | 0.067 ** (0.029) | −0.010 (0.031) |
| $W \times$ Time | −0.020 (0.019) | 0.005 (0.000) | −0.019 (0.012) | 0.003 (0.000) |
| rho | 0.435 *** (0.081) | 0.449 *** (0.108) | 0.559 *** (0.071) | 0.527 *** (0.087) |
| sigma2_e | 0.009 *** (0.001) | 0.008 *** (0.001) | 0.004 *** (0.001) | 0.003 *** (0.001) |
| $R^2$ | 0.545 | 0.546 | 0.734 | 0.778 |

Notes: Parenthetical values are t-statistics. ***, **, * represent significance at the 1%, 5%, and 10% levels, respectively. Values in parentheses are standard errors.

**Table 6.** Spatial effects of the logging quota structure on forest carbon sinks.

| Variables | Carbon | | | | | |
|---|---|---|---|---|---|---|
| | $W_1$ | $W_2$ | $W_1$ | $W_2$ | $W_1$ | $W_2$ |
| Quota$_1$ | 0.038 * (0.022) | 0.035 * (0.020) | | | | |
| Quota$_2$ | | | 0.102 * (0.058) | 0.106 * (0.060) | | |

**Table 6.** *Cont.*

| Variables | Carbon | | | | | |
|---|---|---|---|---|---|---|
| | $W_1$ | $W_2$ | $W_1$ | $W_2$ | $W_1$ | $W_2$ |
| Quota$_3$ | | | | | 0.048 ** | 0.049 * |
| | | | | | (0.024) | (0.026) |
| Pgdp | 0.266 | 0.170 | 0.269 | 0.209 | 0.192 | 0.209 |
| | (0.571) | (0.555) | (0.576) | (0.561) | (0.559) | (0.556) |
| Income | 2.418 *** | 3.047 *** | 2.449 *** | 3.096 *** | 2.367 *** | 3.183 *** |
| | (0.717) | (0.749) | (0.729) | (0.763) | (0.664) | (0.766) |
| Urban | 0.030 ** | 0.024 * | 0.029 ** | 0.022 * | 0.032 ** | 0.022 |
| | (0.014) | (0.013) | (0.014) | (0.014) | (0.013) | (0.013) |
| Grain | 0.761 | 0.947 * | 0.679 | 0.796 | 0.680 | 0.818 |
| | (0.551) | (0.541) | (0.548) | (0.536) | (0.532) | (0.515) |
| Reform | 0.470 | 0.169 | 0.314 | 0.054 | 0.276 | 0.109 |
| | (0.514) | (0.474) | (0.527) | (0.488) | (0.504) | (0.471) |
| Closure | 0.046 | 0.424 | 0.025 | 0.379 | 0.107 | 0.153 |
| | (0.000) | (0.000) | (0.000) | (0.529) | (0.000) | (0.530) |
| Time | 0.024 | −0.006 | 0.034 | 0.003 | 0.042 | 0.016 |
| | (0.000) | (0.236) | (0.000) | (0.000) | (0.000) | (0.245) |
| $W \times$ Quota | 0.010 | 0.043 | 0.069 * | 0.107 * | 0.154 *** | 0.232 *** |
| | (0.032) | (0.043) | (0.104) | (0.101) | (0.046) | (0.081) |
| $W \times$ Pgdp | −0.622 | −1.570 | −0.650 | −1.680 * | −0.869 | −1.476 |
| | (0.831) | (1.039) | (0.837) | (1.049) | (0.711) | (1.039) |
| $W \times$ Income | −0.869 | −3.322 *** | −2.609 *** | −3.330 *** | −2.531 *** | −3.424 *** |
| | (0.711) | (0.820) | (0.764) | (0.833) | (0.701) | (0.829) |
| $W \times$ Urban | −0.020 | 0.017 | −0.023 | 0.008 | −0.190 | 0.008 |
| | (0.021) | (0.024) | (0.021) | (0.024) | (0.915) | (0.024) |
| $W \times$ Grain | 0.027 | −0.038 | −0.070 | −0.054 | 0.164 | −0.088 |
| | (0.982) | (0.822) | (0.971) | (0.824) | (0.558) | (0.803) |
| $W \times$ Reform | −0.723 | 0.193 | −0.653 | 0.144 | −0.173 | −0.002 |
| | (0.818) | (0.943) | (0.827) | (0.966) | (0.572) | (0.943) |
| $W \times$ Closure | 0.072 | 0.110 | 0.039 | 0.091 | 0.231 | 0.118 |
| | (0.467) | (0.524) | (0.474) | (0.000) | (0.331) | (0.000) |
| $W \times$ Time | 0.079 | −0.131 | 0.101 | −0.080 | 0.314 | 0.090 |
| | (0.197) | (0.000) | (0.201) | (0.241) | (0.000) | (0.000) |
| rho | 0.055 ** | 0.005 * | 0.066 ** | 0.034 * | 0.024 ** | 0.061 * |
| | (0.111) | (0.178) | (0.109) | (0.173) | (0.106) | (0.177) |
| sigma2_e | 1.056 *** | 1.018 *** | 1.081 *** | 1.042 *** | 1.055 *** | 1.032 *** |
| | (0.124) | (0.021) | (0.132) | (0.125) | (0.113) | (0.125) |
| $R^2$ | 0.186 | 0.201 | 0.193 | 0.204 | 0.241 | 0.244 |
| Direct effect | 0.039 * | 0.035 * | 0.105 * | 0.109 * | 0.050 ** | 0.048 * |
| | (0.022) | (0.020) | (0.059) | (0.061) | (0.024) | (0.027) |
| Indirect effect | 0.006 | 0.042 | 0.079 * | 0.111 * | 0.158 *** | 0.220 *** |
| | (0.036) | (0.043) | (0.114) | (0.159) | (0.047) | (0.084) |
| Total effect | 0.045 * | 0.077 ** | 0.184 * | 0.220 ** | 0.208 *** | 0.268 *** |
| | (0.030) | (0.038) | (0.103) | (0.142) | (0.045) | (0.077) |

Notes: Parenthetical values are t-statistics. ***, **, * represent significance at the 1%, 5%, and 10% levels, respectively. Values in parentheses are standard errors.

On the other hand, the sub-forest region state and collective ownership are China's two primary forms of forestland ownership [49]. Statistics from the ninth NFI show that the

state holds 95.4% of the forestland and that 92.1% is natural forest. However, in the south, which includes Zhejiang, Anhui, Fujian, Jiangxi, Hubei, Hunan, Guangdong, Guangxi, Guizhou, and Hainan, >90% of the forests are collectively owned and devolved to rural households for management and use [12]. As a result, we independently re-estimated the impact of the logging quota for the collective forests in the southern and non-southern regions (Table 7).

**Table 7.** Spatial heterogeneity of the effects of the logging quotas on forest carbon sinks.

| Variables | Carbon | | | |
|---|---|---|---|---|
| | $W_1$ | $W_2$ | $W_1$ | $W_2$ |
| Quota | 0.032 *** | 0.031 *** | 0.038 ** | 0.037 *** |
| | (0.006) | (0.005) | (0.015) | (0.014) |
| Pgdp | 0.255 | 0.011 | 0.286 | 0.295 |
| | (0.360) | (0.343) | (0.809) | (0.785) |
| Income | −0.176 | −0.180 | 2.296 *** | 4.038 *** |
| | (0.568) | (0.474) | (0.835) | (0.975) |
| Urban | −0.004 | 0.009 | 0.040 ** | 0.043 ** |
| | (0.008) | (0.008) | (0.019) | (0.019) |
| Grain | 0.573 | 0.179 | 0.402 | 0.549 |
| | (0.377) | (0.334) | (0.736) | (0.695) |
| Reform | 0.044 | 0.158 | 0.190 | 0.137 |
| | (0.289) | (0.275) | (0.727) | (0.717) |
| Closure | 0.114 | 0.211 | −0.175 | −0.145 |
| | (0.279) | (0.400) | (1.727) | (0.502) |
| Time | 0.002 | 0.027 | −0.037 | −0.039 |
| | (0.000) | (0.012) | (0.436) | (0.524) |
| $W \times$ Quota | 0.020 | 0.022 | 0.057 * | 0.115 ** |
| | (0.015) | (0.025) | (0.024) | (0.052) |
| $W \times$ Pgdp | −1.448 *** | −2.644 *** | −0.214 | −1.494 |
| | (0.558) | (0.763) | (1.063) | (1.293) |
| $W \times$ Income | −0.187 | −0.096 | −2.174 ** | −4.030 *** |
| | (0.607) | (0.531) | (0.927) | (1.049) |
| $W \times$ Urban | −0.006 | −0.031 ** | −0.016 | 0.065 * |
| | (0.015) | (0.016) | (0.034) | (0.037) |
| $W \times$ Grain | −0.533 | −0.821 | −0.609 | 0.288 |
| | (0.763) | (0.825) | (1.325) | (0.974) |
| $W \times$ Reform | 0.090 | 0.109 | −0.516 | −0.176 |
| | (0.357) | (0.369) | (1.188) | (0.915) |
| $W \times$ Closure | 0.564 * | 1.000 *** | −0.235 | 0.060 |
| | (0.250) | (0.279) | (1.923) | (0.000) |
| $W \times$ Time | −0.027 | −0.082 | 0.212 | 0.156 |
| | (0.097) | (0.114) | (0.501) | (0.001) |
| rho | 0.053 ** | 0.438 * | 0.002 * | 0.098 ** |
| | (0.151) | (0.269) | (0.1333) | (0.205) |
| sigma2_e | 0.081 *** | 0.069 *** | 1.563 *** | 1.437 *** |
| | (0.015) | (0.013) | (0.207) | (0.202) |
| $R^2$ | 0.623 | 0.642 | 0.222 | 0.280 |
| Direct effect | 0.036 ** | 0.031 ** | 0.039 ** | 0.035 ** |
| | (0.007) | (0.006) | (0.016) | (0.016) |

**Table 7.** *Cont.*

| Variables | Carbon | | | |
|---|---|---|---|---|
| | $W_1$ | $W_2$ | $W_1$ | $W_2$ |
| Indirect effect | 0.031 (0.020) | 0.007 (0.018) | 0.053 * (0.037) | 0.110 ** (0.055) |
| Total effect | 0.067 ** (0.024) | 0.037 ** (0.019) | 0.091 ** (0.040) | 0.145 *** (0.055) |
| Southern collective forest region | Yes | | No | |
| Non-southern collective forest region | No | | Yes | |

Notes: Parenthetical values are t-statistics. ***, **, * represent significance at the 1%, 5%, and 10% levels, respectively. Values in parentheses are standard errors.

The findings, which are shown in Table 6, demonstrate that each logging quota significantly affects the forest's carbon sinks. According to the spatial effect decomposition method, the non-southern collective forest region has a significantly higher overall impact from the logging quota on forest carbon sinks, with a significantly positive indirect effect, compared to the southern collective forest region, where the indirect effect is not significant. Due to the non-southern collective forest region covering most of the country in China, reducing the logging quota significantly impacts social investment in forestry. It is associated with a higher expectation of instability. As a result, it has a spatial spillover effect and is inimical to the sustainable regeneration of afforestation and the expansion of forest carbon sinks. However, the northern state-owned forest zone is primarily positioned for ecological conservation, so even if the logging quota in the southern collective forest region is lowered, the influence on logging in the nearby areas is minimal, making the spatial spillover effect less visible.

## 6. Discussion

### 6.1. The Logging Quota Scheme and Forest Recovery

The logging quota scheme was created to protect forests, but once it is implemented, it may have inevitable, unavoidable negative repercussions. Commonly referred to as collective forestland HRS, this system kept established forests under collective ownership while arranging for equitable reallocations of these forests for household management [19]. The ownership, disposal rights, and usufruct of the trees have been returned to rural households with collective forest tenure reform advancement. The reform improved individual tenure by devolving the centralized land management responsibility of the collectives to rural households. However, the logging quota now sharply limits the timber harvest opportunity on household-managed lands, and the continued imposition of logging quotas offsets much of the improved incentive derived from improved household rights [18]. As a result, this contradiction has become more glaring in the latest round of reform. The logging quota undermines the advantages of collective forest tenure reform by discouraging rural families from reforestation and reforestation. In reality, the logging quota that was implemented to preserve forest stock has worked as a deterrent in forestry management [51].

Additionally, the State Forestry Administration's (SFA) centralized quota setting and allocation system permits rent-seeking on the part of solid local elites who regulate timber harvest and trade by issuing logging permits and who also receive the largest share of benefits [26]. It is asserted that prohibiting rural households from harvesting and commerce deters them from investing in long-term forest protection investments. Therefore, extensive afforestation and reforestation projects are primarily responsible for China's forest recovery and growth of forest carbon sinks [12]. The logging quota scheme has been in effect for over 35 years and is anticipated to last for many years as an essential strategy for limiting timber removal and fostering forest restoration in China. This necessitates a program that

gradually eliminates the logging quota scheme so that rural households can benefit from the forest.

### 6.2. Tradeoff between Forest Carbon Sinks and Timber Production

Although logging reduces the effectiveness of forests as carbon sinks, the notion that commercial and environmental uses should be arbitrarily separated, creating two distinct classes of forests worldwide, is illogical and absurd. Classifying the management of forests with a commercial and environmental focus makes sense. However, China's current classified management system needs to be fixed, as once a forest is designated for ecological purposes, regular commercial activities are severely restricted or even illegal under Chinese law [18]. It is essential to manage forestland for a variety of functions. Understory farming and forest tourism are both possible uses for forest land that is used for ecological objectives. Furthermore, it is not necessarily true that the advantages for the environment are associated with the commercial use of forestland [58]. Even though logging has remained stable in recent decades, other measures to tighten longing control, such as reserving commercial forests for public use, have been taken. Governments have transferred some timberlands for ecological protection [18]. More recently, to preserve ecosystems, all silvicultural activities in naturally regenerated forests have been prohibited again [59]. Globally, various policy tools are increasingly being used to save forests. Comparatively, overlapping policies harm forest conservation [53]. Overlapping forest conservation policies (i.e., the logging quota and nature reserve and national forest park policies) may in fact have moved forest harvesting from areas with one form of protection to another. Besides, commercial activity is almost entirely restricted in a forest area that has been designated as a nature reserve or national forest park where there the logging quota is already in place. This would exacerbate conflicts between nearby towns and nature reserves, harming efforts to conserve forests.

Notably, once a forest is designated for ecological purposes, normal commercial activities are strictly regulated or prohibited by law. Following a restriction on log production and a sharp rise in timber consumption, the gap between China's wood supply and demand has been widening quickly in recent years [60]. China is now the world's top wood importer since importing from abroad has become the primary remedy for timber scarcity [61]. Fearnside et al. [62] made the case that the Brazilian Amazonian deforestation resulted from China's rising import demand for timber. A large portion of China's imported wood is obtained illegally, impacting the ecosystem worldwide [63]. All of this necessitates immediate action by Chinese politicians to ease restrictions on timber harvesting to guarantee that most of China's long-term wood supply originates from local sources.

### 6.3. Limitations

Several restrictions on this study could affect the findings. First, we have heavily relied on government figures for logging quota and forest stock in general, as well as data collected at the provincial level. Following an audit of the total amount of logging, the State Forestry Administration submits a report to the State Council for approval before gradually distributing quotas to the provinces, municipalities, and counties [49]. The State Council reviews the annual logging quotas every five years, and China also undertakes an inventory of its forest resources every five years. Regrettably, the official quantity of logging and forest stock is only available at the provincial level rather than at a smaller scale. The province-level study utilized here is relevant because the alternative livelihood effect of the logging quota may be seen more clearly on a large scale than on a small one. At the provincial level, changes in forest species composition and rural labor movements are more apparent. Furthermore, these constraints might be overcome in future research thanks to the growing accessibility of high-resolution, remotely sensed imagery and sophisticated land cover discrimination techniques. Second, we could not determine how the logging quota might affect social investment in forestry because there are no statistics on this topic. As a result, we cannot further investigate the mechanism underlying the effect of the

logging quota on the forest carbon sinks. Third, more explanatory variables need to be considered and investigated in future studies due to the restricted access to a broader range of relevant variables. Given that these elements may significantly impact forest carbon sinks, we should pay closer attention to natural predictors like terrain, temperature, rainfall, and other similar variables.

## 7. Conclusions

China's forest cover expanded from 8.6% to 20.36%, creating the world's largest artificial forest, covering 61.69 million hectares [64]. Due to the government's efforts, notably, the logging quota scheme, several significant initiatives for large-scale forestation and forest conservation have been implemented. Even though there are numerous opposing, contentious viewpoints, the scheme has rarely been the subject of careful, let alone thorough, examination [65]. Although it was intended to prevent forest degradation and even help the forest stock recover, it has instead discouraged forest management and further weakened the incentives for planting timber forests, resulting in a stagnant state of forest quality. Its impact on forest carbon sinks is, therefore, still unknown. Using data from 29 provinces collected over 30 years, we designed and assessed an empirical model to address this problem and separate the effects of the logging quota on changes in forest carbon sinks.

In order to accurately assess the effects of the logging quota on forest carbon sinks, it was necessary to (1) capture spatial spillovers; (2) differentiate between socioeconomic factors in order to match them coherently with the logging quota and forest carbon sinks; and (3) differentiate between the likely endogeneity brought on by the potential autocorrelation of forest carbon sinks. By overcoming these obstacles, we gathered a collection of conclusions. Over the sample observation period, there was a significant local clustering in the spatial autocorrelation of forest carbon sinks. The logging quota showed considerable spatial spillover effects favorable to establishing forest carbon sinks and beneficial to the growth of forest carbon sinks in the surrounding areas. Rural households will explore alternate sources of income in the context of logging control, such as time changes in forest species or migration to cities for job seeking. We also explored the influence of logging quota on rural labor migration and the percentage of timber forest afforestation areas.

The findings show no strong correlation between logging quotas and rural labor migration but that logging quotas considerably impact the proportion of timber forest afforestation area across the whole period of our data coverage. The empirical findings above confirm that rural households need help to organize timber production freely to maximize their profits. According to the aforementioned empirical findings, the logging quota decreases the timber forest afforestation area percentage. Additionally, the logging quota needs to incentivize rural labor migration, which could encourage workers to abandon managing forestland. Spatial heterogeneity affects how logging quotas affect forest carbon sinks. A positive spatial spillover effect of increasing the tending logging quota or the logging quota in the non-collective forest zones is beneficial to encouraging the formation of forest carbon sinks in the surrounding areas. While tending logging enhances stand quality and boosts forest carbon storage, principal logging may result in carbon leakage. In summary, the primary finding of this study is that deregulating logging, particularly tending logging, could encourage the expansion of local forest carbon sinks and make it easier to expand forest carbon sinks in neighboring regions.

**Author Contributions:** Conceptualization, Z.Z.; methodology, Z.Z. and W.Z.; validation, Z.Z. and M.H.; data curation, Z.Z.; writing—original draft preparation, Z.Z. and J.H.; writing—review and editing, Z.Z. and M.H.; visualization, Z.Z.; supervision, W.Z.; project administration, Z.Z. All authors have read and agreed to the published version of the manuscript.

**Funding:** This research was funded by the Humanities and Social Sciences Youth Foundation, Ministry of Education of the People's Republic of China (No. 21XJC790015) and the National Natural Science Fund of China (No. 72003069).

**Institutional Review Board Statement:** Not applicable.

**Informed Consent Statement:** Not applicable.

**Data Availability Statement:** We entered into a usage agreement with National Forestry and Grassland Science Data Center (NFGSDC) before we could use the data, which came from NFGSDC. The agreement states that the data could not be made public.

**Acknowledgments:** We thank the National Forestry and Grassland Science Data Center (NFGSDC) for providing the National Forest Inventory (NFI) data to carry out the study.

**Conflicts of Interest:** The authors declare no conflict of interest.

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
