# Peer review of "Is Regulation Protection? Forest Logging Quota Impact on Forest Carbon Sinks in China"

_sustainability, doi:10.3390/su151813740_

Round 1

Reviewer 1 Report

Dear authors, 

I suggest reducing the amount of tables if possible. 

Please consider following comments: 

line 33: I wouldn't start  the article using "outpace C losses". maybe another sentence and in the second line you put carbone (C) so that everybody knows it's carbon. 

line 487: why would they want to change the forest composition? 

line 496: which is harmful to the resilience of forest ecosystems. needs citation 

line 496: this is very questionable, you are not taking into account social effects of such displacement. I would erase this paragraph. 

line 507: "to encourage rural households to migrate" this is based on the authors subjective opions and not on a deduction of the data. 

line 537: erase the . after region

line 564: the restrictions of your study go at the end of the discussion section

line 572: what do you mean by: "Any discrepancies in the statistics could cause our analyses to be inaccurate"? totally inaccurate? than I wonder if they are worth publishing them? 

line 43: expressed in a strange way, improve English

line 69: . after citation. 

line 160: a space is lacking

line 171: "Statistics on forests and tree crops were derived from the findings of the six national 170 forest inventories conducted between 1989 and 2018. Second, the National Forest Inven- 171 tory (NFI)". It is not clear to me what the difference between the national forest inventories and the national forest inventories are. 

line 184: "Everything is related to everything else, "  I would erase this. 

line 187: what do you mean by spatial spillover effects? 

lines 196: I am not sure if it is necessary to put the formulae, as I consider this index is well known. also the explanation is not necessary. you can put the range of values, which would be enough.

formula 11: I would directly put the simplified one (with explanation). 

line 252: explain what IPPC is

line 313: what is GDP? 

line 317: explain the concept of r environmental Kuznets curves (EKC) 

line 360: explain better the epigraph: Table 1. Variable definition and descriptive statistics. used for....

line 361: what is the source of this graphic? 

line 369: Figure 2. Growth trends for forests as carbon sinks and logging quota. idem

line 418: possible explanation

lines 422-423: at the same time, they are more cognizant of the need to protect forests" how do you explain this? or how did this happen?

line 592: I am not sure what you mean by: , the inconsistency between the logging quota and rural households' operation of the forestland or 593 trees becomes apparent." how is this becoming apparent= maybe you need a citation or explain better. 

line 613: "is illogical and absurd". interesting, but needs justification. 

line 619: "Comparatively, overlapping policies harm forest conservation" explain this better please. 

finally, I would shorten the conclusion to two or three paragraphs max. 

kind regards. 

English is good, needs some minor revisions. 

Author Response

We appreciate your help in revising the work. We have updated the material in accordance with your recommendations. Attachments is a description of the particular modification and the corresponding revisions in the re-submitted files.

Reviewer 2 Report

Conceptual analysis has to be replaced by "Theoretical framework". The introductory section is a way too long. Divide it into a brief introduction and a "Literature review". Employing correlation analysis is fine. However, then replace the notion of dependent variable by an outcone variable. Furthermore enrich the literature on the role of environmental policy stringency because the paper fails to incorporate serious works. Refer to  https://doi.org/10.3390/su12093880

Furthermore, the paper needs a comprehensive proofreading. For the sake of robustness the author could augment the empirican part by a simple OLS.

Author Response

We appreciate your help in revising the work. We have updated the material in accordance with your recommendations. The attachment is a description of the particular modification and the corresponding revisions in the re-submitted files.

Round 2

Reviewer 2 Report

The paper has been substantially improved and can be published now.